# Self-Perceived Impact of COVID-19 Pandemic by Dental Students in Bucharest

**DOI:** 10.3390/ijerph18105249

**Published:** 2021-05-14

**Authors:** Laura Iosif, Ana Maria Cristina Ţâncu, Andreea Cristiana Didilescu, Marina Imre, Bogdan Mihai Gălbinașu, Radu Ilinca

**Affiliations:** 1Department of Complete Denture, Faculty of Dental Medicine, Carol Davila University of Medicine and Pharmacy, 17–21 Calea Plevnei Street, Sector 1, 010221 Bucharest, Romania; laura.iosif@umfcd.ro (L.I.); marina.imre@umfcd.ro (M.I.); 2Department of Embryology, Faculty of Dental Medicine, Carol Davila University of Medicine and Pharmacy, 17–21 Calea Plevnei Street, Sector 1, 010221 Bucharest, Romania; andreea.didilescu@umfcd.ro; 3Department of Dental Prosthesis Technology and Dental Materials, Faculty of Dental Medicine, Carol Davila University of Medicine and Pharmacy, 17–21 Calea Plevnei Street, Sector 1, 010221 Bucharest, Romania; 4Department of Biophysics, Faculty of Dental Medicine, Carol Davila University of Medicine and Pharmacy, 17–21 Calea Plevnei Street, Sector 1, 010221 Bucharest, Romania; radu.ilinca@umfcd.ro

**Keywords:** dental education, COVID-19 pandemic, student perception, psychological impact, educational impact, online learning, student wellbeing

## Abstract

All social and economic systems worldwide, including the educational one have been disrupted by escalating the global COVID-19 pandemic. One of the most impacted areas were the medical and dental education fields, due to the forced break from clinical practice during the lockdown, which affected both the educational part, as well as the patients. Thus, the main goal of our research was to investigate the impact of the COVID-19 pandemic on the dental students’ education as related to their perceptions and evaluations, in Carol Davila University of Medicine and Pharmacy, Bucharest, Romania. A cross-sectional study was conducted on 878 dental students who reported their perception of the psychological and educational impact of this period by completing a Google Forms questionnaire. Collected data were statistically analyzed using Stata/IC 16. There was a severe psychological impact among the respondents, the levels of stress being perceived as high and very high (33.83%, *n* = 297; 28.59%, *n* = 251), similar to high and very high anxiety feelings (26.54%, *n* = 233; 24.26%, *n* = 213). Very high educational impact from the point of view of the acquisition of practical skills (48.52%, *n* = 426) and future professional perspectives (38.95%, *n* = 342) were recorded. While online theoretical learning ability was principally low (37.93%, *n* = 333) despite consistently modified time allocated to the individual study (44.35%, *n* = 389), most of the students evaluated the efficiency of lecturers in online courses as neutral (41.12%, *n* = 361). New dentistry teaching programs will have to be adopted taking into account the dynamics of the pandemic and its strong impact on our students, in order to improve both their wellbeing and the sustainability of dental education.

## 1. Introduction

The most severe health crisis that the 21st century has faced so far has its origins in a zoonotic virus from the Coronaviridae family. Along with its predecessors, which have also become human pathogens, namely Severe Acute Respiratory Syndrome Coronavirus and Middle East Respiratory Syndrome Coronavirus (SARS-CoV-1 and MERS-CoV), the novel Coronavirus (SARS-CoV-2) belongs to the positive-sense single-stranded enveloped ribovirus family [1] with primary respiratory tropism. However, SARS-CoV-2 differentiates itself from the rest of the aforementioned viruses through an unprecedented pathogenicity and contagiousness, even in a lower mortality rate context [2]. Its aggressiveness, which is considerably higher than that of other long existing human pathogen beta coronaviruses (e.g., HCoV-229E, HCoV-OC43, HCoV-NL63, and HCoV-HKU1) [3], including its above-mentioned predecessors, is first and foremost defined by its unique clinicopathological features. The worst manifestations requiring ICU management to support life [4] include pulmonary complications [5], cardiac complications [6], neurological complications [7], and, last but not least, renal complications [8]. Despite accumulated knowledge until present, the Coronavirus disease 2019 (COVID-19) clinical manifestations and long-term sequelae continue to form an incomplete puzzle. However, from a medical point of view, it is certain that the pathological picture has to be understood as a multi system disease [9].

From a transmission point of view, this is primarily realized through the Flügge droplets [10], through infectious saliva or secretions, through nasal or oral contact with contaminated hands [11] and to a lesser extent via aerosols [12]. Further, the capacity of the novel Coronavirus to be easily spread even by asymptomatic carriers represents another challenge in effectively combating the spread of the disease [13].

These aspects explain the gravity of the data presented by the WHO (World Health Organization) after more than a year since the start of the pandemic, namely the death of 3,003,794 people and the infecting of 140,322,903 individuals (18 April 2021). The SARS-CoV-2 pandemic is far from being drawn, if we strictly take into consideration the destruction of global health. Thus, the two month since the diagnosis of the first individual and the spread across the globe, determined the WHO officials on 30 January 2020, to declare the COVID-19 epidemic as a public health emergency of international concern, with some countries already instating the first lockdown measures.

Similar to other pandemics in the history of humanity (the Bubonic Plague, the Spanish flu, SARS-CoV-1, Ebola, Influenza A (H1N1) several turbulences were created by the severity of the COVID-19, emphasizing the many already existing problems and challenges in the health sector [14,15,16,17]. In this regard, the COVID-19 outbreak has hit with unexpected tsunami-like force the population across the globe, with only a few states—arguably—claiming themselves to be COVID free (e.g., Turkmenistan, North Korea, a number of Pacific nations) since the start of the pandemic and until the present day. Even the most advanced medical systems have been severely affected by the unprecedented number and gravity of cases, while conventional healthcare services are being compromised in order to meet the demands of caring for COVID-19 patients. The unusual spread of the novel Coronavirus has claimed several victims including among frontline healthcare workers (HCW) who are fighting against the pandemic. Thus, up until recently the median of the HCW deaths in 100,000 per population of the country was 0.05, according to data from a study on 27 countries including Romania [18].

The challenges, created not only by COVID-19 but also the global quarantine, effect the wellbeing of all social groups in the affected countries [19,20]. Thus, the quarantine strategy has its side effects including the general economical consensus warning of a global recession [21] caused by this black swan event, as the COVID-19 crisis is currently called [22]. Despite convergent global government efforts, primary sectors such as agriculture, oil and petroleum division, secondary sectors involving exports, imports, manufacturing businesses, and tertiary sectors including all service provision industries, such as financial industry, transport, travel, hotels, and restaurants, along with all levels of the education system, from preschool to tertiary education [22] have been short-, medium- and long-term dramatically affected. 

As far as the education system is concerned, in order to reduce the speed of infection, different countries have introduced various and dynamically modified policies, ranging from complete closure of educational institutions in states such as Mexico, Venezuela, Brazil, Ecuador, Bolivia, Poland, Bulgaria, Romania, Turkey, Greece, to partial closure in Sweden, Czechia, Slovakia, Bosnia and Herzegovina, Estonia, etc. At the time of our survey, UNESCO estimated that over 300 million learners have been affected by the closure of educational institutions in a total of 28 countries worldwide, which consisted in 17.8% of the total enrolled learners across the globe [23].

Thus, the functionality of all educational systems was overwhelmed by the COVID-19 pandemic, with some of them already being faulty and vulnerable in certain areas, deepening the already existing faults even further. Although the physical closure of educational institutions (schools, colleges, universities, and other educational institutions) helped significantly reduce the spread of SARS-CoV-2 it has also led to challenges for both students and teachers [24] which involved changed communication channels for lecturers and administrative support, new assessment methods, different workloads and performance levels [14,25,26,27]. When it comes to universities, challenges included, on one hand, the elimination of all interior and outdoor campus activities, such as exams, workshops, study in the libraries, conferences, festive events, meetings with university colleagues, sports, and, on the other hand, the rapid and often unprepared implementation of distance learning solutions [28] on e-platforms. Apart from internet connectivity issues in some countries and regions across the globe, the switch from face-to face learning to e-learning has revealed problems such as lack of information technology, educational materials, and digital technology skills [29], as well as emotional, mental, and physical health breakdowns, due to dramatic changes in one’s social life (e.g., moving back home due to the closure of university dorms, lack of traveling, meetings and parties, etc.) [14,30,31,32,33], personal financial situation of students (e.g., loss of student job, concerns about their financial situation) [20,34], and COVID-19 infections within this category. However, the positive aspect is that although COVID-19 is a threat to humanity, it has also forced academic institutions to best deliver theoretical online content, such as redesigned courses, problem-based learning tutorials, theoretical and practical exams [35] via well-known platforms such as email, Google^®^ educational tools, Skype^®^, Facebook^®^, Instagram^®^, YouTube^®^, WhatsApp^®^, Telegram^®^, and others [36]. Despite these new positive educational adaptations, the faculties of medicine are a special case as practice cannot be replaced by theory especially when it comes to clinical curricula.

One of the most impacted fields was dental medicine, exhibiting disastrous consequences on final year students [37] caused by the lack of clinical practice, due to the great risk of dentistry practitioners of contracting the coronavirus. In this regard, both patients who could no longer benefit from treatments in faculties’ clinics, as well as many dental schools worldwide suffered the consequences caused by this major disruption. Most of the dental schools across the globe, including Carol Davila University of Medicine and Pharmacy in Bucharest, Faculty of Dental Medicine, had to switch from the usual, face-to-face way of teaching, to the online one and later to the hybrid way, in the attempt to slow down the virus spread. 

Although students are young and, thereby, generally not in any of the specific high risk groups of coronavirus infection, due to the serious health consequences it might bring, they are still a category that has experienced dramatic effects during the first and second wave of the COVID-19 pandemic, beginning with the first months of 2020 and continuing in 2021. Considering that dentistry is primarily a clinical practice [38] profession, dental students are perhaps even more affected in their immediate and distant future [24] regarding various unlearned manual and cognitive skills, so are necessary to the profession. There are several studies on dental students’ positive approach and extensive use of e-learning methods, but research on their perceptions regarding the effects of the pandemic on their clinical practical activity, learning motivation and impact on the dental profession in the future is limited.

Therefore, the main goal of this paper was to investigate the educational, professional, and emotional impact of the COVID-19 pandemic on the life of the dental students in the Faculty of Dental Medicine in the Carol Davila University of Medicine and Pharmacy in Bucharest, Romania. Further objectives of our research are the improvement of the educational system both on short- and long-term through finding the limitations, advantages and disadvantages of online learning, highlighting the issues dental students in Bucharest are forced to face as well as recognizing some vulnerable groups in this category.

## 2. Materials and Methods

### 2.1. Study Design and Sampling Procedures

A cross-sectional analytical study was conducted on a sample of dental students attending the Carol Davila University of Medicine and Pharmacy in Bucharest, Romania, in order to investigate the psychological problems and perception related to the theoretical and practical achievement of the academic curriculum experienced during the COVID-19 pandemic. Therefore, all students (*n* = 1808) studying at the Faculty of Dental Medicine of Bucharest, Carol Davila University of Medicine and Pharmacy were invited via institutional email on 5 December 2020, to participate anonymously in completing a questionnaire. A number of 878 students responded to the questionnaire, which represents 48.56% of the students enrolled in the Faculty of Dental Medicine in Bucharest in the 2020–2021 academic year. The mean age was 21.46 years old (SD 2.42; range 18 to 43) (Q1). Regarding the sex distribution, among them 684 were females (77.9%) (Q2). Students’ distribution per year of study was as follows: 1st year, *n* = 191 (21.75%); 2nd year, *n* = 208 (23.69%); 3rd year, *n* = 103 (11.73%); 4th year, *n* = 138 (15.72%); 5th year, *n* = 115 (13.1%); 6th year, *n* = 123 (14.01%) (Q3). Living alone was reported by 160 students (18.22%) (Q4). Living alone (Q4) was reported by 160 students (18.22%).

At the beginning of October 2020, the Faculty of Dental Medicine in Bucharest reopened after a fully online educational semester program, developing the clinical training activities in a hybrid system, namely both with the physical presence of students, as well as further in the online system.

### 2.2. Data Collection and Ethical Considerations

The data were collected using a Google Forms questionnaire, which was distributed through the League of Dental Students in Bucharest, in order to ensure the impartiality of the answers, via email on December 5th, as previously mentioned. Then followed 2 reminders on December 8th and December 11th. The questionnaire was available for one week 24/24, and the window period for responses closed on December 12th. During the data collection timeframe, no notifications regarding the technical functionality of the questionnaire were registered.

Before answering the questions, the respondents were informed that their participation in the questionnaire survey was voluntary and anonymous. The study protocol was approved by the Ethics Commission of the Scientific Research of the Carol Davila University of Medicine and Pharmacy in Bucharest, Romania, with the corresponding ethical approval no. 31215/27.11.2020. Questionnaire submission by the students was considered as informed consent from their side to participate in this study. No incentives were used for study participation.

### 2.3. Survey Instrument

Since no validated tool could be found in the extant literature, the authors developed a survey in order t assess the pandemic’s effect on Romanian dental students with the goal of developing educational strategies with immediate applicability. The questionnaire was constructed in Romanian for the present study, and then translated into English. For the English version, a pilot study was conducted on 10 PhD dental medicine students to test the questionnaire, in order to ensure both the understanding of the meaning of the questions and the accuracy of their translation into English. The questions, their used abbreviations, the respective five-answer scale, as well as the Romanian translation, can be accessed in the Appendix A.

Thus, the questionnaire comprised 3 sections: the first section consisted of socio-demographic data (Age—Q1; Gender—Q2; Year of study—Q3; Leave alone—Q4), the second consisted of 4 items regarding the self-reported psychological impact of COVID-19 pandemic, respectively, the last section included a set of 5 items referring to the educational impact felt during this period. The survey covered some aspects such as anxiety, stress level, motivation for individual study, as well as student concerns regarding the online learning system, respectively, the impact on academic performance and clinical skills acquisition during the COVID-19 outbreak.

Each question within Section 2 and Section 3 had a five-scale response organized according to the Likert scale model, thus containing 5 variants of closed answers. In the case of these 9 questions, the construction of the answers was not identical, so that each question was attached to the response scale with the corresponding coding. Additionally, regarding these two sections of questions and to make sure that student answers focused on the pandemic but not pre-pandemic circumstances, survey questions related to educational concerns and psychological impacts most included key words, respectively, phrases such as “COVID-19 pandemic”, ”SARS-CoV-2”, or “this period” in their structure.

Students’ psychological impact of the COVID-19 pandemic was evaluated by questions such as “How do you assess the level of stress you feel as a result of the COVID-19 pandemic?” (Q5), “How do you assess the anxiety feeling related to a possible infection with SarsCov2”? (Q6) and “How do you assess the change in sleep quality during the COVID-19 pandemic?” (Q7). The response categories for these 3 questions were identical, respectively, (1) Non-existent, (2) Very low, (3) Low, (4) High, and (5) Very high, so that, regarding the first three questions, the higher the score the stronger the psychological impact on students. The last question in this section, namely “How do you appreciate the motivation for individual study compared to the period before the COVID-19 pandemic?” (Q8) had the following categories of answers: (1) No affect, (2) Minor affect, (3) Neutral, (4) Moderate affect, and (5) Majoraffect, with the same direct proportionality of the score compared to the level of impact perceived by the students, as before. Additionally, regarding the questions in this section (Q5–Q8), it is worth mentioning that the scale of answers conduct from positive options to negative options, and that this ordering was thought to avoid the common tendency of left-side bias for our survey-takers.

Further, the first question stated in the educational section was “How do you appreciate your own level of acquiring of academic information through the online teaching system?” (Q9), the answers going in this case from negative to positive, in order to avoid the symmetric direction of the survey responses. The response categories were therefore: (1) Non-existent, (2) Very low, (3) Low, (4) High, and (5) Very high, so that regarding this item, the higher the score, the better the learning ability of students through online methods. Efficiency of university teachers was formulated as “How do you evaluate the efficiency of teachers in transmitting theoretical and practical information in online teaching courses?” (Q10) and the scale of responses included the following variants: (1) Very low, (2) Low, (3) Neutral, (4) High, and (5) Very high. In this situation, also, the selection of a response with a higher score was correlated with a higher performance among university teachers, the direction of the answers being also from negative to positive, as in the previous question. The next question in the educational section, namely “How do you think the time allocated to the individual study has modified in your case, compared to the period before the COVID-19 pandemic?” (Q11) was assigned the following scale of answers: (1) Not modified, (2) Minor modified, (3) Neutral, (4) Moderate modified, and (5) Major modified. The choice of an answer variant with a higher score reflected only the magnitude of the change in the individual study time, the authors not requesting the specification related to its increase or minimization (The choice of an answer variant with a higher score reflected the extent of the change in the individual study time, the authors not requesting the specification related to its increase or minimization). The fourth question of the section, respectively “In light of the COVID-19 pandemic, how much was your acquisition of practical skills regarding dental work affected, given that you normally would have gained them from patients during traditional clinical activities?” (Q12) was joined by a scale of answers of the type (1) No affect, (2) Minor affect, (3) Neutral, (4) Moderate affect, and (5) Major affect. Additionally, in this case the perceived impact was even greater the higher the numerical value of the response. Finally, the professional perspective of the future was assessed by the question “How do you evaluate the impact of this period on the profession of dentist in the future?” (Q13) with response alternatives (1) Non-existent, (2) Very low, (3) Low, (4) High, and (5) Very high. In this regard, as in the case of the two previous questions (Q11 and Q12), the higher the answer value, the more negative the students’ perception, the direction of the scale of responses returning thus from positive to negative.

### 2.4. Data Management and Analysis

Students’ answers were automatically collected into Google Forms. A data sheet was generated in Microsoft Excel (Microsoft Corporation, Redmont, Washington, WA, USA), in which the variables were coded. Data were thereafter transferred and analyzed using Stata/IC 16 (StataCorp LLC, College Station, TX, USA).

Data distributions were expressed as means, standard deviations, and percentages, appropriately. Possible correlations were tested using the Spearman’s rank correlation coefficient.

A *p*-value less than 0.05 was considered statistically significant.

## 3. Results

Regarding the psychological impact reported by the students, as indicated by the descriptive analysis for this section (Table 1 and Table 2), although high stress and anxiety levels were experienced during the COVID-19 pandemic, minor changes in sleep quality were mostly reported, and the individual study motivation remained in the majority of cases unmodified.

Furthermore, the descriptive analysis of the educational impact (Table 3 and Table 4) highlighted a low level of acquiring of academic information through the online teaching (Q9), although the students’ opinion regarding efficiency of university lecturers (Q10) was rather neutral, just as in the case of the individual study time (Q11). However, the majority of students reported maximum impact on gaining practical skills (Q12) and future professional perspective (Q13).

A significant positive correlation coefficient was computed between anxiety and level of stress (rho = 0.51, *n* = 878, *p* < 0.0001). Significant positive correlations were also detected between sleep quality and anxiety (rho = 0.33, *n* = 878, *p* < 0.0001), as well as between sleep quality and level of stress (rho = 0.42, *n* = 878, *p* < 0.0001).

Academic learning was positively correlated with academic teaching (rho = 0.59, *n* = 878, *p* < 0.0001), and time for individual study (rho = 0.17, *n* = 878, *p* < 0.0001), but negatively correlated with gain of practical skills (rho = −0.21, *n* = 878, *p* < 0.0001) and future professional perspective (rho = −0.23, *n* = 878, *p* < 0.0001). Negative correlations were also recorded between academic teaching and time for individual study (rho = −0.14, *n* = 878, *p* < 0.0001), gain of practical skills (rho = −0.16, *n* = 878, *p* < 0.0001), and future professional perspective, respectively (rho = −0.14, *n* = 878, *p* < 0.0001). Gain of practical skills was positively correlated with future professional perspective (rho = 0.54, *n* = 878, *p* < 0.0001).

No significant correlations were found between the ranks of study motivation and the years of study. Although the year of study did not significantly impact the answers to the second section of questions regarding the perception towards education, (Figure 1, Figure 2 and Figure 3), some important tendencies can be observed. There was an increase in the number of those significantly affected with regards to the gain of practical skills (Figure 4), and their professional future, with professional perspectives worsening with the advancement in clinical study years (Figure 5).

## 4. Discussion

Despite public healthcare and technological progress made in the two decades of the 21st century, the COVID-19 outbreak has hit worldwide medical systems due to lockdown measures. Current clinical activity has been significantly modified with the students of the faculties of medicine being suddenly impacted by the lack of practical activity, lack of acquiring critical thinking skills regarding appropriate therapeutic decisions, etc. Moreover, due the high risk of contracting COVID-19 that dentistry professionals asses [23,39,40], starting 8 March 2020, all Faculties of Dental Medicine in Romania have decided one by one to interrupt physical lecture attendance, clinical and preclinical practical activities, as well as exams until the end of the second semester, with dental schools activities being limited to emergency treatment by faculty. Furthermore, the state of emergency announced on March 16th, through a Military Ordinance, among other restrictions, involved the complete closure of dental offices, with only a few selected dental care centers remaining open for emergencies in each city.

In the majority of Romanian dental medicine faculties, the second academic semester started using a hybrid system, with lectures being held online, while clinical practical activity was held partially physically on-site. Nevertheless, due to an increase in COVID-19 cases in Bucharest and the predict of a second wave, starting October 2020, the Faculty of Dental Medicine in the Carol Davila University of Medicine and Pharmacy in Bucharest was forced to adopt an atypical curriculum, with the exception of usual practical activities performed on patients. These were replaced with simulating the clinical maneuvers demonstrated by the lecturers, followed by student practice in groups with fewer individuals, using simulation models, study models, extracted teeth, prosthetic appliances and casts, or training on adjustable articulators, etc., according to each clinical discipline.

Due to this significant challenge in dental education, our study aims to both evaluate the psychological impact of the pandemic, as well as the impact of online theoretical and practical learning on students, following a period of time which was long enough to objectively draw conclusions about this completely new academic model for us. In this regard, it is mandatory to specify that within the Faculty of Dental Medicine in Bucharest e-learning due online applications or software platforms before the pandemic have never been used.

Thus, up until now several reports were made on the impact of the pandemic on students’ emotional health looking at anxiety disorders, resentments, depression, anger, mental fatigue [14,20,31,32], etc., among them. All of these add to the competitiveness, rigor and increased demands and to the early responsibility when treating patients [41], which characterize dental medicine education in general, and which place it in one of the most stressful academic environments [42,43], while also being one of the most emotionally and psychologically demanding professions. Our study highlights the significant psychological impact on students, as both stress and anxiety levels are directly correlated and, therefore, perceived by the majority of them as being elevated, due to the supplementary psychological stress triggered by the pandemic. According to this study, the percentage of students who significantly feel the stress after 10 months into the pandemic, is even higher than that of those who are afraid of contracting the novel coronavirus, regardless of their study year. The same student stress impact was highlighted by Hakami, Z. et al. [44], in a study performed through the evaluation of DASS-21 (Depression, Anxiety, and Stress Scale) scores, at the beginning of the pandemic. Nevertheless, it is worth mentioning the current scarcity of comparative data from other European countries on pandemic related stress, among the same study sample.

Regarding the anxiety level, results divergent to our study have been reported by Lingawi, H.S. et al. [45], the authors revealing rather mild anxiety scores among the 258 dental medicine students who participated in the study, with the research being done only a few months after the pandemic had begun and during the summer holiday. Thus the reduced fear of contracting SARS-CoV-2, was, therefore, most probably correlated with the complete physical absence of students in the academic activities both theoretical and practical. In a similar manner, a study on a large number of medical students (7143) in China [46], informed on lower percentages of moderate (2.7%) to severe anxiety (0.9%) among them, the motivation most probably being the significant attention from the central Chinese government regarding the COVID-19 outbreak and its influence on students’ psychological health, thus strongly guiding the educational sector in the area of psychological counselling, psychological assistance hotlines [47], etc.

Several studies have reported a relationship between study motivation and sleep disorders [48]. Nevertheless, although sleep quality in our participants was at a low point at the time the study was conducted, the individual study motivation has not dropped, but it was described as unaffected instead. It is, however, known that uninterrupted sleep may optimize learning capacity and motivation [49]. From a psychological point of view, the lowering of interest in studying is, thus, barely noticeable, with the result of our study remaining a scientific subject open to debate.

Regarding the actual educational aspect, literature is abundant in information on the perception of dental medicine students during the COVID-19 pandemic. In this regard, one of the most surprising and maybe concerning findings of our study was that, despite unmodified study motivation, the vast majority of dental students from Bucharest, regardless of their respective study year, had difficulties in acquiring academic information through the online teaching system. This was most likely due to the fact that this academic model was absolutely inexperienced at our university until the time of the pandemic. Similar or even worse perceptions regarding the efficiency of academic online learning have been reported in Pakistan dental schools [50], Iraqi dental universities [51], and the Faculty of Dentistry University of Indonesia [52]. However, there were some rather positive student opinions according to a study conducted at the Dental School of Justus-Liebig-University Giessen (Germany) [53]. Deficiencies in the online learning process could be explained by the lack of technical computer skills among some students who were unprepared in this regard and forced in this system, the lack of familiarity and virtual learning fatigue [54], increased difficulty to stay in contact with lecturers [55], etc.

Looking further for a reason why our students report a low ability to assimilate theoretical information, while asked if there are potential gaps regarding the online teaching methods among lecturers, it could be observed that their answers were mostly in the neutral zone of the scale. The inclusion of a *neutral* answer option was not randomly chosen at the moment the questionnaire was being constructed as this allowed freedom to the student, an escape button, where a firm opinion was not required, and which allowed them not to choose an answer which did not reflect their true beliefs [56]. Thus, two or even three interpretation options can be discussed, just like the neutral answer was intentionally thought by Likert, where students either *do not know* the answer to the question regarding the lecturers’ efficiency in communicating academic information through the online system (the other answers being: very low, low, high, very high) [57], either they *know but they wish to not respond*, whether they *do not understand the question* [58], or they are *undecided* [59]. Regardless of the interpretation, it is certain that theoretical and practical information communicated by the university lecturers is a subject which needs to be reflected upon and at the same time perfected, as, although students’ perception on the assimilation of online taught curricula could be influenced by multiple factors, it seems that the most important factor is the quality of the courses [60]. Our study certifies this important issue, as indicated by the significant correlation between the efficiency of online teaching by the university lecturers, and the acquisition of academic information by students. On its part, the lecturers’ performance in the online learning process is influenced by their professional ability, which should focus on the interactive pedagogy model through communicative and reflective-active instruction [61], on the online teaching adaptability, as well as on confidence toward utilization of digital learning [52]. The adoption of sustainable models, starting with flipped classroom techniques, where students are invited to review online modules before participating in seminars [62], and continuing with the introduction of haptic technology [63] into the clinical curriculum even after the pandemic, and even the use of novel portable teaching-learning platforms such as DenTeach [64], could be solutions to the issues we are facing.

Furthermore, the individual study time dedicated by each student was for the most participating students, regardless of their respective study year, different from the one they were accustomed to prior to the pandemic, namely *minor modified, moderate modified,* and *major modified*. In this case, it should be mentioned that the question did not intentionally refer to whether it was positively or negatively affected. We did not consider the quantification of this modification to be necessary, but the simple modification of this parameter could be interpreted as a supplementary stress factor for the student, which could consist in another argument regarding the stress level that they reported. This statement is all the more plausible as our study found a significant correlation between the modified individual study time and the low level of acquiring academic information due the online system, which, in turn, is certainly a strong stressor for students. A study conducted by Amir et al. [52], indicates that although students admitted there was more time to study and to review study material during distance learning, one-third of the participants experienced important levels of stress, thus being convergent to our study.

It becomes obvious that the gain of practical skills seemed to cause the greatest concern among our students. This is an unsurprising result, due to the fact that no current teaching method, as demonstrated by the correlational analysis in our study, can substitute the acquisition of manual dexterity and fine motor skills, which are necessary for various clinical procedures, possible only by practicing on real patients. Although it is not a statistically significant result, an increase in the impact on practical skills as the study years progress can be noticed and should be discussed. Thus, first and second year students have reported they are less affected in this regard compared to their colleagues in the last preclinical study year, namely the 3rd one, as well as compared to the students in the clinical semesters, namely from the 4th, 5th, and 6th study years. To the best of our knowledge, till present there are limited studies with a similar assessment. One of these studies, conducted in the only dental school in Malta, suggests that the majority of students were severely impacted by not being able to practice their laboratory or clinical skills for a long period, their manual dexterity skills being, therefore, greatly affected [41].

The final question, which asks the students to predict their dental profession perspective, shows an increased tendency directly proportional to those who consider themselves as vulnerable during the pandemic, as the study year progresses. In our opinion, the reason for this result is the forced break of treatments performed directly on patients by dental students in their final study years, the self-perceived level of acquisition of practical skills being strongly positively correlated to the professional perspective in our study. Regarding the issue of evaluating the dental medicine students’ perception on practicing the profession, literature from the first 6–12 months since the pandemic has been declared by WHO, only presents a few comparative studies. One of these studies, conducted at the University of Otago Faculty of Dentistry in New Zealand [65] and which used a questionnaire since mid-March 2020, reports an increased impact, with almost half of the participating students anticipating a decrease in the clinical competence in their profession. Although the study does not differentiate the impact by study year, its authors suggest that this perception might worsen if clinical activities are suspended for an extended period of time, a hypothesis confirmed by our study, which shows that over 60% of the participants foresee either a high or a very high professional impact. Another opinion, this time from May 2020, but with a similar result to our study, came from Roseman University of Health Sciences (RUHS) College of Dental Medicine, Henderson, NV, USA, [66], which indicates that the majority of students foresee a great negative impact on practicing the dentistry profession from the perspective of employment difficulties on a oversaturated job market, worsened by the COVID-19 pandemic.

Albeit our respondents were in extremely high numbers and equally representing each of the 6 study years, our research has some limitations, such as limited generalizability of sample results, or such as non-investigation of other COVID-19-related impacts on the dental medicine students, namely economic and social factors, cancellation of scientific events, considerations regarding change of profession, and others. Besides, due to voluntary response sampling, it is possible that the respondents were mainly those who had stronger opinions than non-respondents.

Longitudinal studies are necessary to investigate as COVID-19 may more commonly result in general distress, depression, anxiety, and somatization instead of post-traumatic stress disorder. Rather than exacerbate these experiences, some degree of depressive manifestations, short-term adjustment issues, and long-term adaptation to the uncertain future are maybe reasonable or expected responses [67]. These manifestations, also found among our students, indicate the need to promote programs to support good mental health, taking into account some relevant aspects such as the resources to support these measures. Therefore, as resources could be particularly scarce during a serious pandemic situation, as is currently the case, timely psychological support including telemedicine and informal support groups [68] could be a potential help for this young category from our country.

Additionally, what’s clear is that this pandemic has disproportionately impacted vulnerable social groups, such as racial minorities and lower-income families, etc. Thus, stringent research in representative samples is needed to identify the roots of inequities beyond the individual level, also examining community, policy, health care system, and society-level determinants (and their intersections) [69].

However, it is becoming more and more clear that in the following years, perhaps even in the following decade, the measures imposed by the pandemic in one of the most affected cities in Romania, considering the COVID-19 infection rate and mortality rate, will greatly impact the future dentists on both a psychological and a professional level. At the time our report is finished, as the pandemic continues to evolve, forcing the Bucharest dental curriculum to be taught following a hybrid system, it appears that new guidelines and protocols regarding the dental education program will be adopted, according to both this dynamic and our students’ necessities, as it is highlighted by the current study. Moreover, given that the Faculties of Dental Medicine are not only educational and research institutions, but also small hospitals [70], finding sustainable solutions during this period, which are adapted to the local realities as they are showcased by the current study, may contribute to the improvement of both the oral health and wellbeing of the citizens in Bucharest.

## 5. Conclusions

Our research examined the medium-term impact of the COVID-19 pandemic on dental students in Bucharest in respect to their psychological and educational perceptions. To our knowledge, this is one of the few studies currently undertaken on such an extensive number of participants in this field of interest, which further strengthens the accuracy of the results obtained by questioning dental students on their perceptions of the middle period of the pandemic.

Thus, the emotional well-being adversely affected in the case of our students suggests the need to promote good mental health programs among them, along with the development of proactive communication strategies from the university lecturers and the amplification of student involvement into academic plans and decisions. The pandemic impact on dental education is the more destructive, the more the educational process involves the gain of practical skills, which were previously performed by the students directly on the patient.

Furthermore, students’ perceptions regarding the online distribution of the theoretical content of the curriculum may lead to upgrading the e-teaching strategies among our university lecturers. Thus, the results of our study should be seen as a great opportunity to reconsider the clinical activity in the Faculties of Dental Medicine across the country, in the benefit of both the students and the patients, and to optimize the student interaction in an academic environment which will most probably never resemble the one prior to the COVID-19 pandemic or at least for many years to come.

## Figures and Tables

**Figure 1 ijerph-18-05249-f001:**
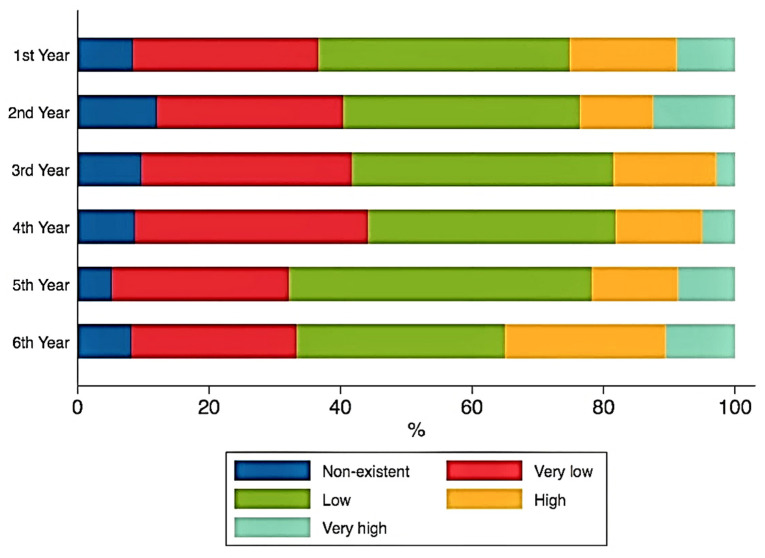
Academic online learning.

**Figure 2 ijerph-18-05249-f002:**
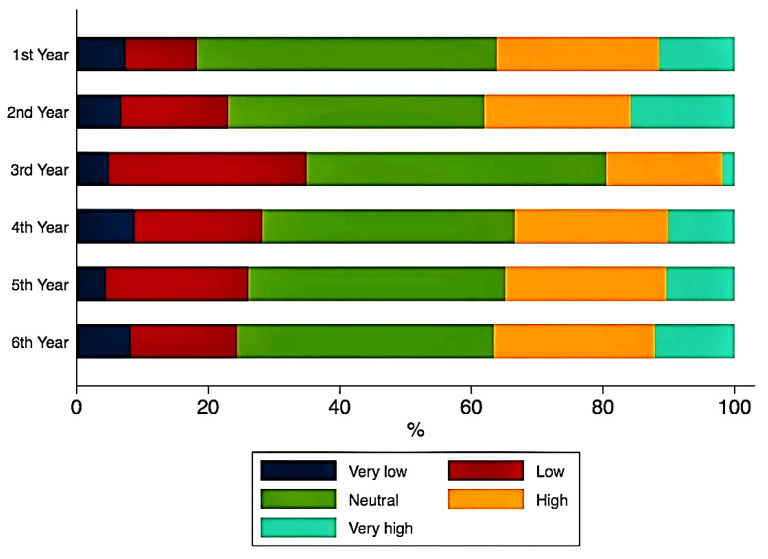
Academic online teaching.

**Figure 3 ijerph-18-05249-f003:**
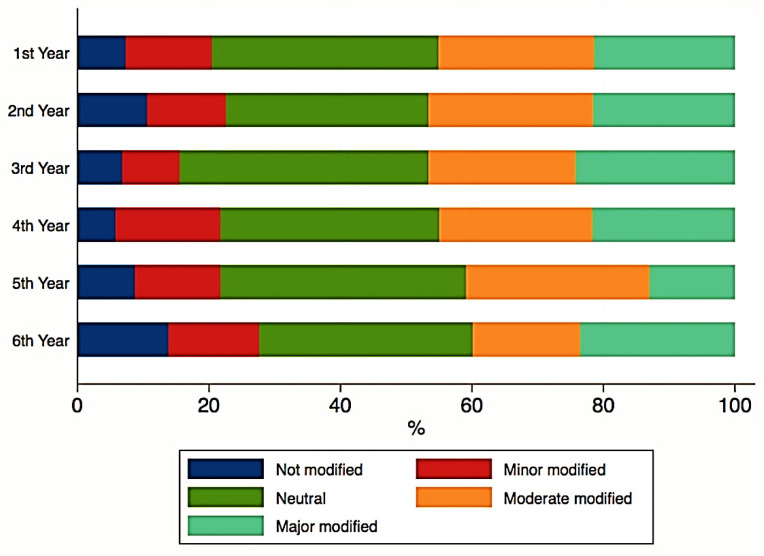
Time for individual study.

**Figure 4 ijerph-18-05249-f004:**
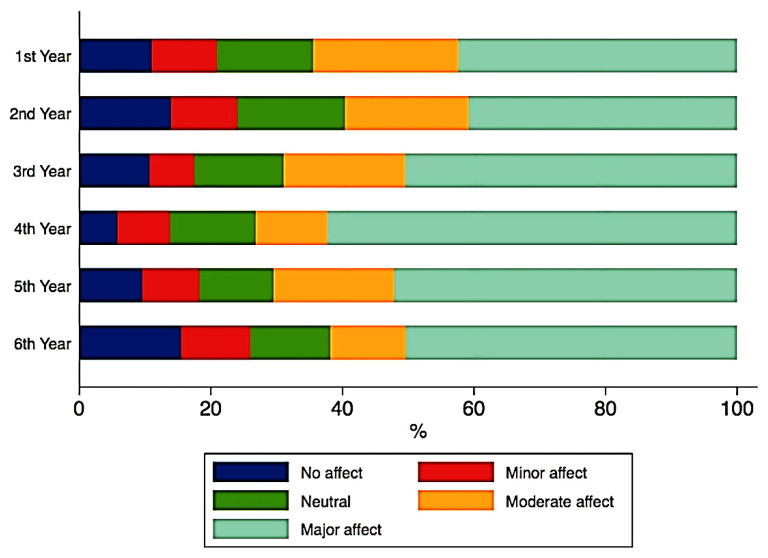
Gain of practical skills.

**Figure 5 ijerph-18-05249-f005:**
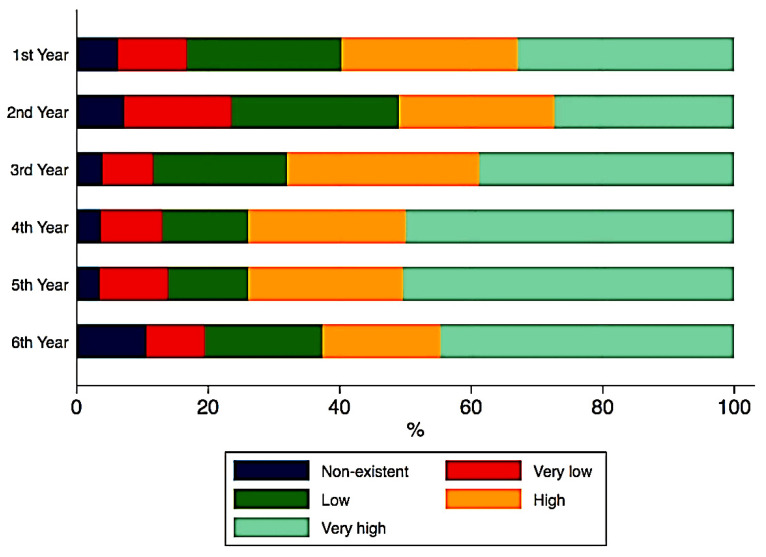
Future professional perspective.

**Table 1 ijerph-18-05249-t001:** Data distribution by responses codes for psychological impact items.

Response Code	Q5. Level of Stress		Q6. Anxiety		Q7. Sleep Quality		Q8. Study Motivation	
	Freq	Percent	Freq	Percent	Freq	Percent	Freq	Percent
1	25	2.85	70	7.97	186	21.18	294	33.49
2	80	9.11	137	15.60	192	21.87	207	23.58
3	225	25.63	225	25.63	238	27.11	212	24.15
4	297	33.83	233	26.54	139	15.83	102	11.62
5	251	28.59	213	24.26	123	14.01	63	7.18

**Table 2 ijerph-18-05249-t002:** Descriptive analysis of responses to psychological impact items.

Questions	Mean (SD)	Median	Mode	Skewness/Kurtosis
Q5. Level of stress	3.76 (1.05)	4	4	−0.57/2.69
Q6. Anxiety	3.43 (1.23)	4	4	−0.34/2.14
Q7. Sleep quality	2.8 (1.32)	3	3	0.19/1.95
Q8. Study motivation	2.35 (1.25)	2	1	0.55/2.27

**Table 3 ijerph-18-05249-t003:** Data distribution by responses codes for educational impact items.

Response Code	Q9. Academic Learning		Q10. Academic Teaching		Q11. Time for Individual Study		Q12. Gain of Practical Skills		Q13. Future Professional Perspective	
	Freq	Percent	Freq	Percent	Freq	Percent	Freq	Percent	Freq	Percent
1	79	9	60	6.83	78	8.88	99	11.28	53	6.04
2	257	29.27	158	18.00	113	12.87	81	9.23	98	11.16
3	333	37.93	361	41.12	298	33.94	122	13.90	173	19.70
4	133	15.15	201	22.89	204	23.23	150	17.08	212	24.15
5	76	8.66	98	11.16	185	21.07	426	48.52	342	38.95

**Table 4 ijerph-18-05249-t004:** Descriptive analysis of responses to educational impact items.

Questions	Mean (SD)	Median	Mode	Skewness/Kurtosis
Q9. Academic learning	2.85 (1.06)	3	3	0.28/2.6
Q10. Academic teaching	3.13 (1.05)	3	3	−0.05/2.59
Q11. Time for individual study	3.35 (1.2)	3	3	−0.27/2.29
Q12. Gain of practical skills	3.82 (1.41)	4	5	−0.86/2.35
Q13. Future Professional perspective	3.79 (1.24)	4	5	−0.71/2.43

## Data Availability

The data presented in this study are available from the corresponding authors upon reasonable request.

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
