# Peer review of "Self-Perceived Impact of COVID-19 Pandemic by Dental Students in Bucharest"

_ijerph, 2021, doi:10.3390/ijerph18105249_

Round 1

Reviewer 1 Report

  1. The title requires modification because the article concerns the influence of the pandemic as perceived by the students and not objectively measured.
  2. The introduction sufficiently illustrates COVID-19 consequences for the functioning of the societies, with a detailed discussion of the changes in the education of the dental students.
  3. The description of the procedure and research group is incomplete - there is no information concerning the structure of the research group. These data were included in the research results (chapter 3, line 236-242) but they should be moved to the description of the research group (chapter 2.1). Similarly, the information about the date of the research which is now in chapter 2.2 should be moved to the description of the procedure (chapter 2.1). Summing up, it would be a good idea to include an extensive characteristics of the research group in its description, along with the information concerning its final size, the basic sociodemographic data of the respondents (sex, age, year of studies) and the time of study.
  4. The description of research tools used [chapter 2.3 Survey instruments] requires completing. It is worth giving exemplary statements, and even including the contents of all items comprising each section (apart from the section concerning sociodemographic data, the tool includes 9 items). The information about specific answers assigned to the Likert scale appear  as late as in the discussion of the results. It is thus a good idea to include them in the description of the tool. The identification of psychometric properties of the tool applied is also missing.
  5. In my viewpoint, statistical analyses require significant changes. The Likert scale allows one to determine the extent to which a statement is accepted, thus showing descriptive statistics for the entire research group, such as mean, standard deviation, median, modal value, skewness, kurtosis. Showing the frequency of choosing particular answers seems pointless here.
  6. The results of Spearman’s rho correlation are incomplete and they are described superficially; they require expanding.
  7. The information included in the beginning of the discussion of the results (lines 285-333) functions more like the introduction to the research in terms of content – they should be moved to chapter 1 (‘Introduction’).
  8. The conclusions formulated on the basis of such modest statistical analyses are invalid to a large extent. In the discussion the meaning of the issue undertaken is well-explained and the results of similar research are shown.
  9. The conclusions are well-formulated.

Reviewer 2 Report

This study examined the effects of the pandemic on the educational and psychological well-being in dental students.  The authors developed their instrument with specific items addressing the pandemic, and their response rate was very good.  This study was well conducted, and however specific hypotheses were not presented.  This was an online survey study aimed at description rather than inference.

Reviewer 3 Report

Although the topic is current and of interest, the manuscript requires major revisions for content and language. Specific comments: 1. "The surprising contagiousness of SARS-CoV-2 compared to its predecessor consists in its firstly abundantly replication in upper respiratory epithelia of the nasopharynx and oropharynx, and is thus efficiently transmitted [10], while SARS-CoV-1 primarily targets pneumocytes and lung macrophages in lower respiratory tract tissues [11]. From a transmission point of view, this is primarily realized through the Flügge droplets [12], through infectious saliva or secretions, through touching one’s nose or mouth with contaminated hands [13] and to a lesser extent via aerosols [14], as is the case with all respiratory viruses. The capacity of the novel coronavirus to be easily spread even by asymptomatic carriers represents another Achile’s Heel in front of all global public health and control programmes towards fighting the spread of the disease [15]" - I am not sure what is the direct relevance of this verbose paragraph. The introduction is overly lengthy and should be made more concise and succinct. 2. "Achilles' Heel" was misspelled. 3. Were the students incentivized to participate in the study? 4. Were duplicate responses excluded from the analysis? How did the authors guard against duplicate responses, e.g. IP filtering? 5. "Despite enormous medical, public healthcare and technological progress in the early 21st century" - were progress only made in the early 21st century? 6. When presenting results, please provide the exact n and accompanying percentage, e.g. "very high (33.83%, n=?; 28.59%, n=?), similar to high and very high anxiety feelings (26.54%, n=?; 24.26%, n=?), despite mostly unaffected study motivation (33.59%, n=?)." 7. There is a paucity of statistical testing to substantiate the current conclusions. 8. How much self-directed or e-learning did the university do before and during the pandemic? What was the change? This was unclear. 9. Rather than over-pathologize these experiences, some degree of sadness, anxiety, fear, anger, paranoia, and short-term adjustment issues and long-term adaptation to the uncertain future are perhaps reasonable or expected responses. The majority of mental challenges following COVID-19 may be “reactive” in nature. It may be in response to the fear and stress of contagion, especially given the possibility of asymptomatic spreaders in the community. It may be the consequence of hospitalization for infected individuals or strict measures to curb and contain the pandemic, with “lockdown” living, loss of livelihood, and financial hardship (citation: pubmed.ncbi.nlm.nih.gov/32943541). Longitudinal studies are necessary to investigate this. What's clear is that this pandemic has disproportionately impacted racial minorities and lower-income families (citation: pubmed.ncbi.nlm.nih.gov/32391864). 10. There was no data availability statement.

Round 2

Reviewer 1 Report

I accept a new version of the article.

Reviewer 3 Report

Thank you for the revisions.

Specific comments:

  1. "... have collapsed similar to a domino effect under the pressures of the present major health crisis of the 21st century" - please avoid such superfluous language. Suggest to change to something more direct and succinct, e.g. "have been disrupted by the escalating global pandemic".
  2. As resources could be particularly scarce during a serious pandemic situation, timely psychological support could also take many forms, including telemedicine and informal support groups (citation: ncbi.nlm.nih.gov/pmc/articles/PMC7405627).
  3. Please rephrase "... the severely affected emotional status of our students suggests the need to amplify psychological counselling among them".
